# A new method based on melatonin-mediated seed germination to quickly remove pesticide residues and improve the nutritional quality of contaminated grains

Lingyun Li[1☯], Baoyan Li[1☯], Henghua Qu[2☯], Shan Tian[3]*, Zimeng Xu[3], Lulu Zhao[3], Xueqin Li[3], Baoyou Liu[1]*

**1** Yantai Academy of Agricultural Sciences, Yantai, Shandong, China, **2** Yantai Agricultural Technology Extension Center, Yantai, Shandong, China, **3** Life Science College, Luoyang Normal University, Luoyang, Henan, China

☯ These authors contributed equally to this work.
* shanshantian2018@163.com (ST); ytnkyzbs@126.com (BL)

**Data Availability Statement:** All relevant data are within the manuscript.

## Abstract

In the present study, we attempted to use melatonin combined with germination treatment to remove pesticide residues from contaminated grains. High levels of pesticide residues were detected in soybean seeds after soaking with chlorothalonil (10 mM) and malathion (1 mM) for 2 hours. Treatment with 50 μM melatonin for 5 days completely removed the pesticide residues, while in the control group, only 61–71% of pesticide residues were removed from soybean sprouts. Compared with the control, melatonin treatment for 7 days further increased the content of ascorbic acid (by 48–66%), total phenolics (by 52–68%), isoflavones (by 22–34%), the total antioxidant capacity (by 37–40%), and the accumulated levels of unsaturated fatty acids ($C_{18:1}$, $C_{18:2}$, and $C_{18:3}$) (by 17–30%) in soybean sprouts. Moreover, melatonin treatment further increased the accumulation of ten components of phenols and isoflavones in soybean sprouts relative to those in the control. The ability of melatonin to accelerate the degradation of pesticide residues and promote the accumulation of antioxidant metabolites might be related to its ability to trigger the glutathione detoxification system in soybean sprouts. Melatonin promoted glutathione synthesis (by 49–139%) and elevated the activities of glutathione-S-transferase (by 24–78%) and glutathione reductase (by 38–61%). In summary, we report a new method in which combined treatment by melatonin and germination rapidly degrades pesticide residues in contaminated grains and improves the nutritional quality of food.

## Introduction

Modern agricultural practices involve the wide use of pesticides, which pose a threat to human health through various routes, such as residues in food [1–4]. Consequently, no human population groups are completely unexposed to pesticides [5]. Excessive preharvest spraying with pesticides can lead to high levels of pesticide residues in plant tissues and crop

**Funding:** This study was supported by grants from the Major Basic Research Project of Natural Science Foundation of Shandong Province (grant number ZR2022ZD23), the Innovation Team Fund for the Fruit Industry of Modern Agricultural Technology System in Shandong Province (grant numbers SDAIT-06-12, SDAIT-06-11), the Key Project of Shandong Natural Science Foundation (grant number ZR2020KC026), the Key R&D Plan of Shandong Province (grant numbers 2021CXGC010802, 2021CXGC010602, 2022CXGC02070912), the Key R&D Plan of Yantai City (grant numbers 2021NYNC015, 2022XCZX094 to Baoyou Liu) and State Key Laboratory of Soil Erosion and Dryland Farming on the Loess Plateau, Chinese Academy of Sciences (grant number 190412 to Shan Tian). The sponsors or funders DONOT play any role in the study design, data collection and analysis, decision to publish, or preparation of the manuscript.

**Competing interests:** The authors have declared that no competing interests exist.

seeds [6,7]. In addition, pesticides are frequently applied to stored crop seeds to prevent pest infestation and pathogen infection [8]. For example, soybean [*Glycine max* (L.) Merrill] seed damage occurs during the post-harvest period, with microbial pathogens or pests causing severe losses of stored soybean seeds [9]. To reduce losses, pesticides are used to prevent pest infestation or pathogen infection [10,11]. However, excessive or unsuitable application of pesticides can result in large amounts of residues being present in stored grains [12–14], which endangers human health [14,15]. Thus, pesticide residues in stored grains must be removed.

Methods have been developed to remove pesticide residues from food plants (e.g., food processing and cooking) [16–18]. However, cooking can severely impair the antioxidant nutrients in food [19]. Recently, certain biodegradation methods have been developed to remove pesticides from plants [20–24]. For example, brassinosteroid application significantly decreases pesticide residues in plants [20]. However, to the best of our knowledge, there are have been no reports on the biodegradation of pesticide residues in grains.

In plants, the pleiotropic molecule, melatonin (*N*-acetyl-5-methoxytryptamine), has various functions, for example, acting as an antioxidant to protect against environmental stresses [25]. Notably, the activities of glutathione S-transferase (GST; EC 2.5.1.18) and glutathione reductase (GR; EC 1.6.4.2), and glutathione (GSH) levels, are increased by melatonin to activate GSH metabolism in plants [26,27]. Interestingly, the activation of GSH, GST, and GR are vital for pesticide removal in plants [22]. For example, melatonin was observed to activate GR and GST activities and reduce chlorothalonil and malathion residues in post-harvest jujube fruit [24].

Soybean is an important source of food for humans and feed for animals, containing abundant proteins, lipids, carbohydrates, and various secondary metabolites [28]. Interestingly, it does not contain a high level of GSH [29], which can be used prevent and treat numerous disorders [30,31]. The seed reserves can be degraded and used to synthesize new cell components, especially antioxidant metabolites, in the developing embryo after germination [32,33]. In general, soybean sprouts can provide humans with sufficient health-promoting phytochemicals and can be prepared within 5–7 days after initial germination [34–36]. For example, germination can produce high levels of antioxidant nutrients, such as vitamin C, GSH, isoflavones, and phenolics in legume seeds and sprouts [36]. Interestingly, unsaturated fatty acids (especially $C_{18:1}$, $C_{18:2}$, and $C_{18:3}$) not only promote human health [37–39], but also play a role in regulating plant stress responses [40,41]. Compared with that in soybean seeds, germination significantly reduces the unsaturated fatty acid content via lipid metabolism [35]. However, there have been no reports on the effects of germination on the degradation of pesticide residues and the content of unsaturated fatty acids in contaminated grains.

Chlorothalonil and malathion are two widely used high-efficiency and low toxicity pesticides in China. The former is a broad-spectrum fungicide, and the latter is a broad-spectrum insecticide [24]. In the present study, these two pesticides were applied on soybean seeds and the effects of germination, especially combined with melatonin, on pesticide removal and antioxidant nutrient accumulation were investigated. Herein, two problems were addressed. First, how does germination affect pesticide degradation in soybean seeds and sprouts? Second, how does germination affect the antioxidant nutrient (e.g., phenolics, isoflavones, and unsaturated fatty acid) content in pesticide-treated soybean seeds? We hypothesized that germination, especially combined with melatonin, would lead to increased pesticide biodegradation and antioxidant nutrient accumulation in pesticide-contaminated soybean seeds. This work provides a new approach to utilizing pesticide contaminated grains, which can improve food safety and nutritional quality.

## Materials and methods

### Reagent preparation

Melatonin (purity, $\geq$ 98%) was obtained from Macklin Biochemistry and Technique Company (Shanghai, China). Chlorothalonil (purity, $\geq$ 99%) and malathion (purity, $\geq$ 95%) were purchased from Aladdin Reagents (Shanghi) Co., Ltd. (Shanghai, China). Malathion (33 mg) and chlorothalonil (266 mg) were dissolved in approximately 1 mL of acetone, and then diluted to 100 mL with distilled water to achieve the treatment concentration used in the experiment.

### Seed preparation and treatment

The soybean (*Glycine max*) variety used in this study was Kenfeng 16, which was purchased from an agricultural seed company in Luoyang, China. Before the seed germination experiment, we divided healthy and same sized seeds into three groups (control, chlorothalonil, and malathion groups), each comprising about 600 seeds. The seeds were soaked in tap water (containing 1% acetone, v/v), chlorothalonil (10 mM), and malathion (1 mM) for 2 h, respectively. The treated seeds were rinsed in tap water for 5 min, placed at room temperature to dry for 1 h, soaked in tap water for 30 min, and finally transferred into seed germination boxes. For the germination experiment, the 600 seeds in each group were divided into 6 germination boxes (100 seeds per box), with the seeds in 3 germination boxes being sprayed with tap water as a control, and the seeds in the other 3 germination boxes being sprayed with 50 μM melatonin solution as the treatment group. The germination boxes were placed in the dark, at room temperature, with 80% humidity, to germinate for 1 week. On day 7 after treatment, random samples of the sprouting seeds were taken and used to detect the germination rate, germination length, and pesticide residues. The GSH content, and the activities of two enzymes (GR and GST) were determined after treatment for 24 h. On the 7th day, the soybean sprouts were frozen at −70°C for the subsequent detection of ascorbic acid, phenolic content, isoflavones accumulation, and the total antioxidant capacity (Fig 1). The above experiment was repeated 2–3 times, and the measured results were used for later data analysis.

### Seed germination and sprout growth

We tested the germination rate and germination growth of seeds in the germination experiment at 9 a.m. and 9 p.m. every day. When the soybean germination length reached 1 mm, the seeds were considered to have germinated, and we then calculated the number of germinated seeds as a percentage of the total number of seeds (100 seeds per group) as the germination rate [42]. The length of the seed buds was measured using cursor calipers.

### Pesticide assay

The Institute of Agrochemicals Control of the Ministry of Agriculture of China provided the two pesticides used in this experiment (chlorothalonil and malathion). According to the methods reported by Xia et al. [20], the residues of these two pesticides were tested. Specifically, the samples to be tested (soybean seeds or soybean sprouts) were added with anhydrous sodium sulfate (ASS) and petroleum ether (PE), and homogenized evenly using a shredder (12000 rpm, 5 min). The homogenized bean sprouts or seeds were then filtered through a Brinell funnel (preloaded with 10 g of ASS). The filter cake was washed three times with 50 mL of PE each time, and then mixed with the front filter in a flat flask (0.5 L) and dried under a nitrogen flow. The collected pesticide residues were dissolved in PE, and the collected 5 mL solution was detected using gas chromatography (Fuli GC9790; Zhejiang Fuli Analytical Instrument

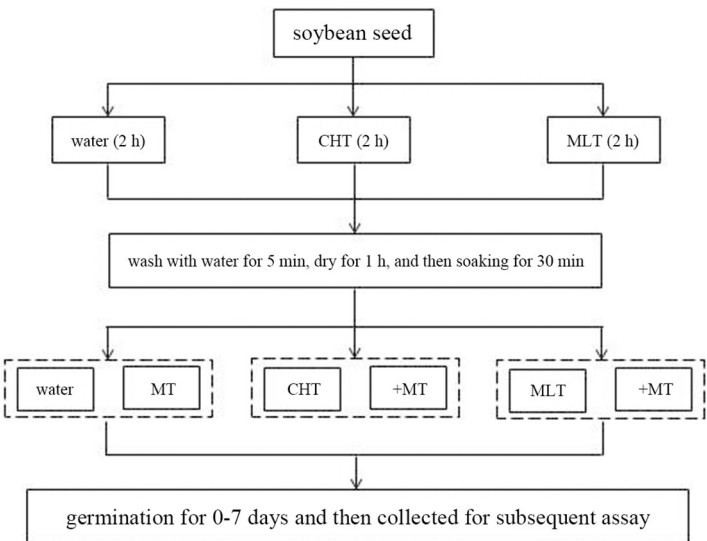

**Fig 1. The flowchart of the study.** This flowchart illustrates how soybean seeds were treated with pesticides and melatonin, as well as how they were grouped. For details, please see the text. CHT, chlorothalonil; MLT, malathion; MT, melatonin.

Co., Ltd., Wenling City, China). The gas chromatograph used was equipped with a phosphorus filter and a flame photometric detector.

## Ascorbic acid and glutathione assay

The content of ascorbic acid was measured quantitatively using the 2,6-dichlorophenol-indoleol (2,6-DPI) titration method [24]. Specifically, 10 g of fresh bean sprouts or soybean seed were mixed with 200 mL of 2% oxalic acid and mashed. The homogenized material was diluted to 1 L using 2% oxalic acid, and then filtered to obtain the filtrate. Then, 10 mL of the ascorbic acid extract filtrate was titrated with 0.01% 2,6-DPI titrant. When the solution turned pink and maintained the color for about 15 seconds, it was judged to have to reached the titration end point. The titration process must be completed in 10–15 minutes. At the same time, a standard curve was generated by titrating ascorbic acid standard solutions with 2,6-DPI.

For the quantitative detection of reduced glutathione (GSH), we used fluorescence spectrophotometry based on phthalic aldehyde as a fluorescent group [43]. In the test, a fluorescence spectrophotometer (Perkin-Elmer LS 55, Waltham, MA, USA) was used to detect and record fluorescence intensity at a wavelength of 420 nm after excitation at 350 nm. At the same time, a standard curve was constructed with different concentrations of GSH standard solutions.

## Analysis of phenolic and isoflavone contents

Soybean samples were prepared for analysis of phenolics and isoflavones using high-performance liquid chromatography according to a previously published method [44]. The instrumentation used for the high-performance liquid chromatography analysis was as described previously [44].

## Total antioxidant capacity (TAC) assay

In this experiment, trivalent iron ion reduction (using the ferric reducing antioxidant power (FRAP) method) was used to detect the total antioxidant ability of the sample [45]. Specifically,

the sample to be tested was homogenized and extracted using 90% methanol. The extract was centrifuged for 5 min at $12,000 \times g$, and the supernatant was collected to detect the total antioxidant ability. Then, 100 μL of the supernatant was added to the reactor, and 2.4 mL of FRAP working solution (containing 0.3 M pH3.6 acetic acid buffer, 10 mM 2,4,6-Tri(2-pyridyl)-s-triazine (TPTZ) solution, and 20 mM ferric trichloride solution; the three were mixed according to a volume ratio of 10:1:1) was added and reacted in a 37°C water bath for 10 minutes, after which the absorption value at 593 nm was recorded. At the same time, a 0.1–2 mM $FeCl_3$ solution was used instead of the sample to construct a standard curve.

### Unsaturated fatty acids assays

Herein, fatty acid ($C_{18:0}$, $C_{18:1}$, $C_{18:2}$ and $C_{18:3}$) extraction and their assays were performed according to the method of Dhakal et al., with some modifications [46]. Firstly, 1% NaOH dissolved in methanol was used to convert the fatty acids into their corresponding methyl esters, which were heated at 55°C for 15 min. Then, 5% methanol hydrochloric acid solution was added to the reaction system, which was heated again at 55°C for 15 minutes. Finally, the fatty acid methyl esters were extracted from hexane and analyzed using a gas chromatograph GC-2010 (Shimadzu, Tokyo, Japan) connected with a GCMS-QP2010 gas chromatograph mass spectrometer (Shimadzu).

### GR and GST activity assays

The sample material to be tested (5 g) was mixed with 50 mL of 50 mM potassium phosphate buffer (containing 1 mM EDTA, 5% polyethylenepyrrole) and mixed evenly at 4°C for 10 minutes using a blender. The resultant slurry was centrifuged at $12000 \times g$ for 20 minutes using a high-speed refrigerated centrifuge, and the supernatant was collected and used to detect the activities of GR and GST. The GR activity was measured at 340 nm using a spectrophotometer according to a previously reported method [47]. The GST activity was also measured using a previously reported method, in which an ultraviolet-visual (UV-Vis) spectrophotometer was used to detect the absorption at 340 nm [24,48].

The Bradford method [49] was used to determine the soluble protein levels, with bovine serum albumin as the standard.

### Analysis of the data

A completely randomized design, comprising 2–3 replicates for each treatment, was used when carrying out the experiments. Duncan's multiple range test in SPSS 13.0 (IBM Corp., Armonk, NY, USA) was used to analyze the data at a significance level of $p < 0.05$.

## Results

### Melatonin improves germination and sprout growth

The inhibitory effect of malathion on soybean seed germination and sprout growth was significantly greater than that of chlorothalonil (Fig 2; $p < 0.05$). Compared with the control, the data showed that after 5 days of treatment with chlorothalonil and malathion, the germination rate of soybean seeds had decreased by 7% and 21%, respectively (Fig 2A; $p < 0.05$). Similarly, malathion reduced the sprout length by approximately 17% compared with the control after sprouting for 7 d (Fig 2B; $p < 0.05$). However, this inhibition of soybean seed germination and sprouting was significantly alleviated by melatonin application (Fig 2; $p < 0.05$). Melatonin treatment increased the 3 d germination rate of chlorothalonil- and malathion-treated soybean seeds by approximately 9% and 32%, respectively (Fig 2A; $p < 0.05$). Similarly, melatonin

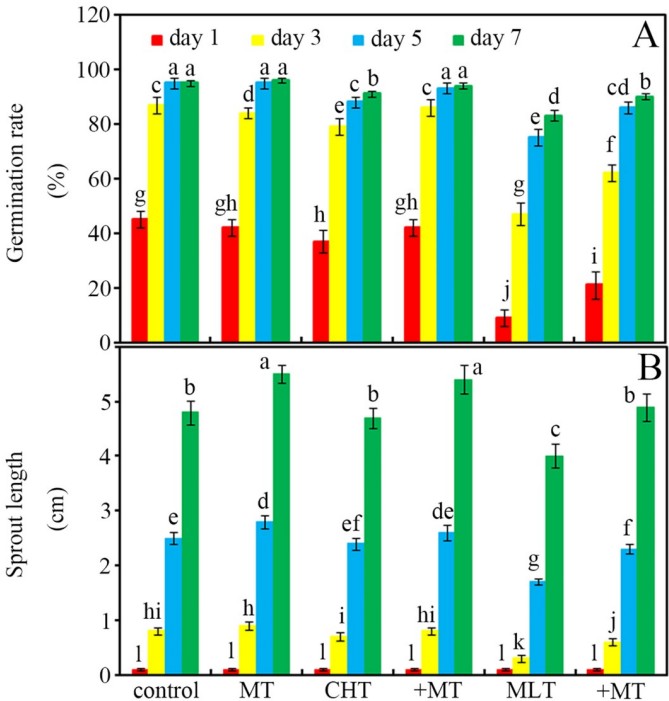

**Fig 2. Melatonin affects seed germination and sprout length.** (A) Melatonin's effects on the germination rate and (B) sprout length were monitored in soybean seeds after imbibition for 7 d. The standard deviation of the mean is indicated by a bar ($n = 3$), and there was no significant difference between means followed by the same letter ($p < 0.05$) within the same time-point. Each group represents 100 soybean seeds or germinated sprouts. CHT, chlorothalonil; MLT, malathion; MT, melatonin.

increased the sprout lengths of the chlorothalonil- and malathion-treated soybean seeds by approximately 14% and 50%, respectively, after treatment for 3 d (Fig 2B; $p < 0.05$).

## Melatonin reduced the levels of pesticide residues

As shown in Table 1, melatonin treatment reduced the levels of pesticide residues in germinated soybean sprouts and seeds. In comparison with dormant seeds, germination decreased the pesticide (chlorothalonil and malathion) residue levels significantly in soybean seeds after imbibition (Table 1). In detail, germination decreased the chlorothalonil and malathion contents by approximately 32% and 23%, respectively, in germinated seeds and sprouts after

**Table 1. Effects of germination on pesticide residues.**

|  | Days after germination and sprouting | | | |
|---|---|---|---|---|
|  | Day 0 | Day 3 | Day 5 | Day 7 |
| **Chlorothalonil** | 157.5 ± 8.2 | 106.7 ± 4.1 | 45.2 ± 5.3 | 4.9 ± 0.4 |
| **+melatonin** | 157.5 ± 8.2 | 14.1 ± 3.6 | nd | nd |
| **Malathion** | 17.3 ± 2.4 | 13.3 ± 1.6 | 6.8 ± 0.1 | 0.8 ± 0.1 |
| **+melatonin** | 17.3 ± 2.4 | 3.7 ± 0.1 | nd | nd |

The effect of melatonin on chlorothalonil and malathion contents (µg g$^{-1}$ dry weight) in pesticide-treated soybean seeds after imbibition for 0, 3, 5, and 7 d under ambient conditions ($n = 2$). nd, not detected.

imbibition for 3 days (Table 1). However, more than 7 days of germination were required for the complete removal of chlorothalonil and malathion residues from germinated soybean seeds and sprouts (Table 1). By contrast, melatonin at 50 μM reduced the chlorothalonil and malathion levels by approximately 91% and 79%, respectively, in germinated seeds and sprouts after imbibition for 3 days (Table 1). Moreover, melatonin could completely remove these two pesticides from germinated seeds and sprouts within 5 days (Table 1).

## Melatonin enhanced the GSH content and related enzyme activities

Melatonin was imbibed into pesticide-treated seeds for 24 h. Thereafter, the content of GSH and activities of two enzymes (GR and GST) were determined (Fig 3). Compared with that in the control group, treatment with chlorothalonil and malathion alone reduced the GSH content in soybean sprouts by approximately 34% and 18%, respectively (Fig 3A; $p < 0.05$). However, compared with that in the seeds treated with chlorothalonil and malathion alone, in the seeds treated with pesticides+melatonin, the GSH content was increased in soybean sprouts by approximately 139% and 49%, respectively (Fig 3A, $p < 0.05$).

Chlorothalonil application enhanced the GR and GST activities by approximately 38% and 85%, respectively, in germinated seeds and sprouts compared with those in the water control (Fig 3B and 3C; $p < 0.05$). By contrast, malathion decreased the GR and GST activities by approximately 27% and 16%, in soybean seeds and sprouts, respectively, relative to those in the water control (Fig 3B and 3C; $p < 0.05$). However, melatonin application further increased the GR and GST activities in germinated seeds and sprouts, regardless of pesticide application (Fig 3B and 3C; $p < 0.05$). In detail, the GR and GST activities increased by approximately 73% and 82%, respectively, in response to melatonin application in germinated soybean seeds and sprouts treated with malathion (Fig 3B and 3C; $p < 0.05$).

## Melatonin enhanced antioxidant nutrient accumulation

Compared with that in the control group, the application of pesticides significantly altered the content of certain antioxidant nutrients (such as ascorbic acid, phenolic substances, and isoflavones) in soybean sprouts (Fig 4; $p < 0.05$). Chlorothalonil and malathion decreased the content of vitamin C by approximately 22% and 16%, respectively, in germinated soybean sprouts and seeds relative to those in the water control (Fig 4A; $p < 0.05$). However, chlorothalonil increased the phenolics and isoflavones contents by approximately 47% and 34%, respectively, in germinated soybean sprouts and seeds relative to those in the water control (Fig 4B and 4C; $p < 0.05$). In addition, chlorothalonil and malathion enhanced the TAC by approximately 31% and 17%, respectively, in germinated soybean sprouts and seeds, relative to that in the water control (Fig 4D; $p < 0.05$). Moreover, melatonin further significantly increased the germination-induced increase in antioxidant nutrient accumulation in germinated seeds, regardless of pesticide treatment (Fig 4; $p < 0.05$). In detail, melatonin increased the ascorbic acid (by ~48%), phenolics (by ~52%), isoflavones (by ~22%), and TAC (by ~38%) levels in malathion-treated soybean seeds and sprout relative to those in the water control after 7 days of imbibition (Fig 4; $p < 0.05$). In addition, similar patterns of change were monitored in water- and chlorothalonil-treated soybean seeds (Fig 4; $p < 0.05$).

As shown in Table 2, pesticides could stimulate the accumulation of certain phenolic substances (e.g., $p$-coumaric acid, ferulic acid, naringin, and hesperidin) and isoflavones (e.g., daidzin, glycidyl, genistein, daidzein, and genistein) in soybean sprouts, and this effect was further enhanced by melatonin ($p < 0.05$). Compared with those in the control group, malathion increased the contents of certain isoflavones in soybean sprouts, such as daidzin, glycine, genistein, daidzein, and genistein, by approximately 21%, 19%, 11%, 20%, and 35%,

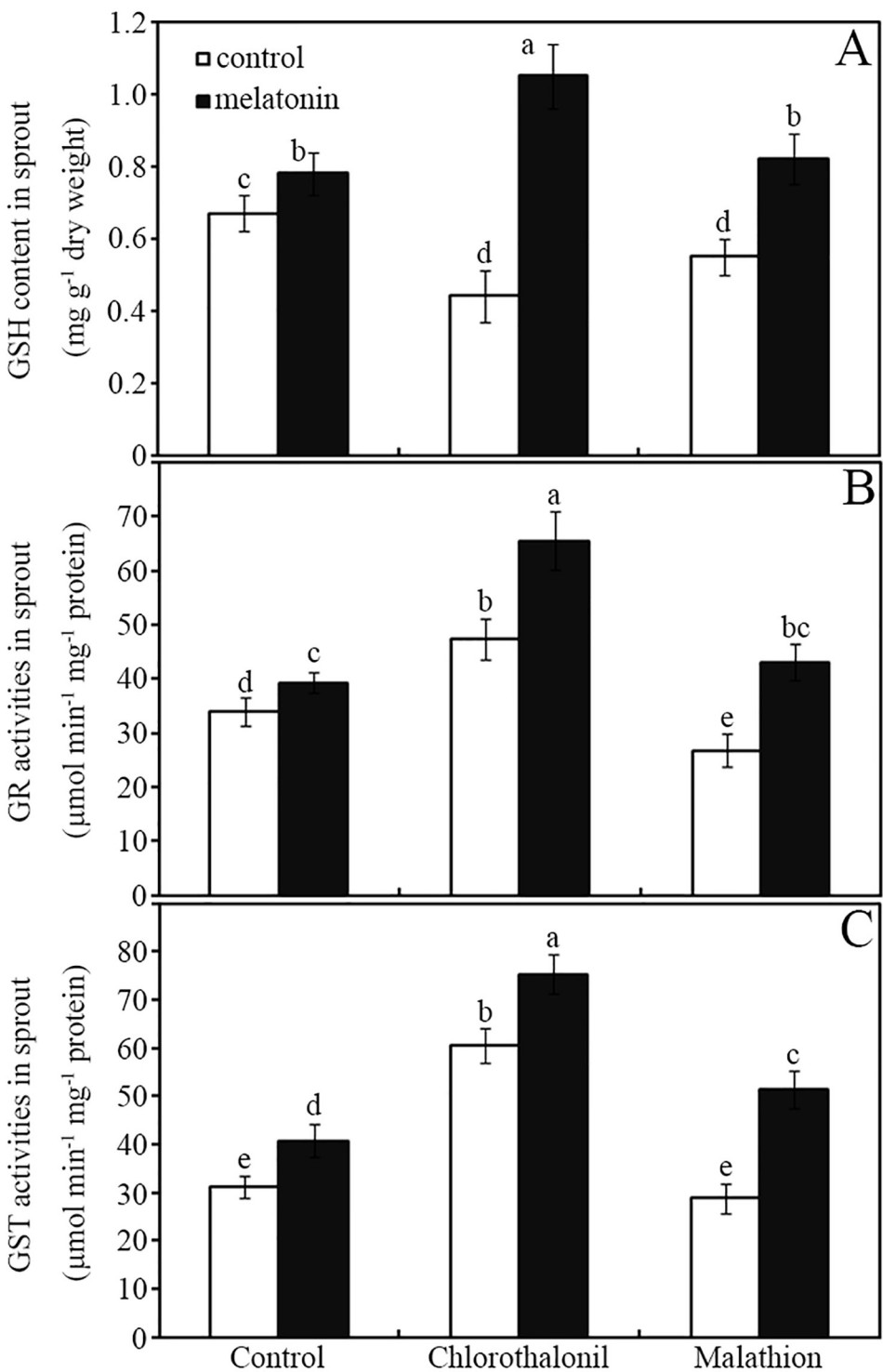

**Fig 3. Melatonin affects the glutathione content and related enzyme activities.** The effects of melatonin on the GSH content (A), and the activities of GR (B), and GST (C) in germinated seeds and sprouts after imbibition for 24 h. The standard deviation of the mean is indicated by a bar ($n = 3$), and there was no significant difference between means followed by the same letter ($p < 0.05$) among the treatments. GR, glutathione reductase; GST, glutathione-S-transferase; GSH, reduced glutathione.

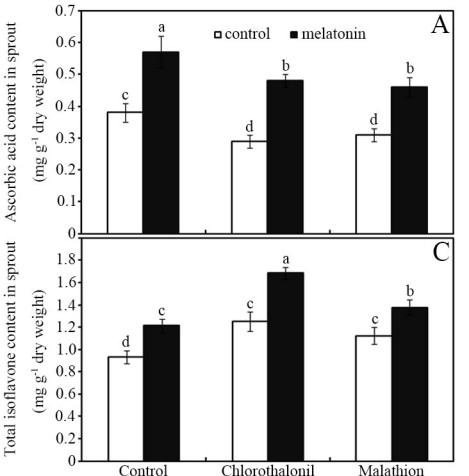
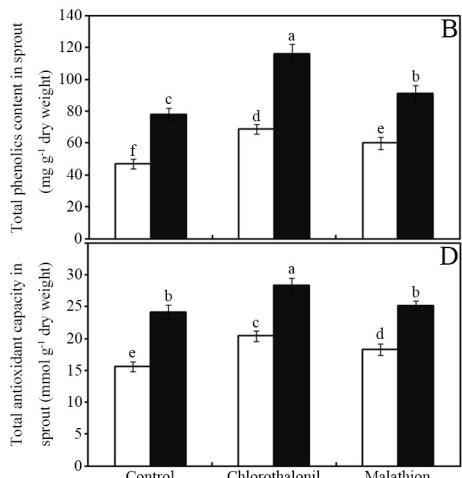

**Fig 4. Melatonin affects antioxidant nutrients.** The effects of melatonin on the accumulation of ascorbic acid (A), the content of total phenolics (B) and total isoflavones (C), and the total antioxidant capacity (D) in soybean sprouts after imbibition for 7 d. The standard deviation of the mean is indicated by a bar ($n = 3$), and there was no significant difference between means followed by the same letter ($p < 0.05$) among the treatments.

respectively (Table 2; $p < 0.05$). Compared with those in the malathion group, the addition of melatonin treatment further increased the content of these isoflavones in soybean sprouts, with daidzin, glycitin, genistin, daidzein, and genistein levels increasing by approximately 11%, 7%, 5%, 8%, and 15%, respectively (Table 2; $p < 0.05$).

## Melatonin enhanced the unsaturated fatty acid content

As shown in Table 3, the accumulation levels of four fatty acids in soybean sprouts were significantly affected by the treatment of seeds with chlorothalonil and malathion. Compared with that in the control, the $C_{18:0}$ content in soybean sprouts increased by about 21% and 47%,

**Table 2. Melatonin affects the content of phenolics and isoflavones and their composition in soybean sprouts.**

|  | Control | MT | CHT | CHT+MT | MLT | MLT+MT |
|---|---|---|---|---|---|---|
| *p*-Coumaric acid | 1.5 ± 0.1[a] | 1.6 ± 0.1[ab] | 1.8 ± 0.1[bc] | 1.9 ± 0.1[c] | 1.5 ± 0.1[a] | 1.7 ± 0.1[b] |
| Ferulic acid | 13.6 ± 0.5[a] | 14.4 ± 0.6[a] | 18.1 ± 0.4[c] | 19.4 ± 0.7[d] | 16.4 ± 0.7[b] | 17.2 ± 0.5[bc] |
| Naringin | 2.6 ± 0.2[a] | 2.8 ± 0.1[a] | 3.5 ± 0.2[c] | 3.9 ± 0.1[d] | 3.1 ± 0.1[b] | 3.3 ± 0.2[bc] |
| Hesperidin | 5.4 ± 0.2[a] | 5.7 ± 0.3[a] | 6.3 ± 0.3[bc] | 6.7 ± 0.2[c] | 6.1 ± 0.3[b] | 6.4 ± 0.2[bc] |
| Salicylic acid | 3.8 ± 0.3[a] | 3.9 ± 0.3[a] | 3.4 ± 0.1[b] | 4.4 ± 0.3[a] | 3.9 ± 0.1[a] | 4.3 ± 0.2[a] |
| ΣTPC (mg kg⁻¹) | 26.9 ± 1.1[a] | 28.4 ± 1.7[a] | 33.1 ± 1.5[b] | 36.5 ± 1.2[c] | 31.0 ± 1.3[b] | 32.9 ± 1.4[b] |
| Daidzin | 180.6 ± 7.7[a] | 199.2 ± 9.1[b] | 244.5 ± 17.5[cd] | 257.8 ± 9.8[d] | 219.4 ± 10.7[c] | 243.5 ± 15.3[cd] |
| Glycitin | 28.6 ± 1.3[a] | 30.8 ± 1.7[a] | 37.5 ± 1.3[c] | 42.8 ± 2.1[d] | 34.1 ± 1.2[b] | 36.4 ± 2.5[bc] |
| Genistin | 274.9 ± 8.7[a] | 297.1 ± 11.2[b] | 338.1 ± 8.7[d] | 342.4 ± 16.5[d] | 304.7 ± 7.7[bc] | 318.7 ± 9.2[c] |
| Daidzein | 65.4 ± 2.4[a] | 71.5 ± 3.2[b] | 87.8 ± 4.1[d] | 95.4 ± 2.9[e] | 78.8 ± 2.1[c] | 85.4 ± 3.4[d] |
| Genistein | 19.7 ± 1.1[a] | 23.1 ± 1.4[b] | 28.6 ± 1.5[a] | 34.3 ± 1.4[e] | 26.6 ± 1.3[c] | 30.5 ± 2.1[d] |
| ΣTFC (mg kg⁻¹) | 569.2 ± 13.2[a] | 621.7 ± 17.7[b] | 726.5 ± 18.8[d] | 772.7 ± 20.5[e] | 663.6 ± 12.2[c] | 714.5 ± 23.4[d] |

Melatonin regulates the total content of phenolics (TPC) and isoflavones (TFC), and the major components of these secondary metabolites in soybean sprouts after treatment for 7 days. Means associated with the same letter are not significantly different ($n = 3$; $p < 0.05$). CHT, chlorothalonil; MLT, malathion; MT, melatonin.

**Table 3. Unsaturated fatty acid accumulation in soybean sprouts.**

|  | $C_{18:0}$ | $C_{18:1}$ | $C_{18:2}$ | $C_{18:3}$ |
|---|---|---|---|---|
| **Control** | $6.2 \pm 0.3^a$ | $27.4 \pm 1.3^a$ | $65.8 \pm 2.1^a$ | $12.6 \pm 1.3^a$ |
| **MT** | $6.4 \pm 0.2^a$ | $27.1 \pm 1.5^a$ | $66.3 \pm 2.6^a$ | $12.4 \pm 0.8^a$ |
| **CHT** | $7.5 \pm 0.3^b$ | $23.6 \pm 1.1^b$ | $53.4 \pm 2.2^c$ | $9.2 \pm 0.6^b$ |
| **CHT+MT** | $6.6 \pm 0.4^a$ | $26.3 \pm 1.4^a$ | $62.5 \pm 2.9^{ab}$ | $11.7 \pm 0.7^a$ |
| **MLT** | $9.1 \pm 0.5^c$ | $20.1 \pm 1.3^c$ | $45.2 \pm 3.7^d$ | $7.5 \pm 0.7^c$ |
| **MLT+MT** | $7.4 \pm 0.3^b$ | $25.4 \pm 1.2^a$ | $58.2 \pm 1.7^b$ | $11.3 \pm 0.5^a$ |

The effects of melatonin on the unsaturated fatty acid content (mg g$^{-1}$ dry weight) in soybean sprouts after treatment for 7 days. Means associated with the same letter are not significantly different for each fatty acid ($n = 3$; $p < 0.05$). CHT, chlorothalonil; MLT, malathion; MT, melatonin.

respectively, after 7 days of treatment of soybean seeds with chlorothalonil and malathion alone (Table 3). However, treatment with chlorothalonil and malathion reduced the content of unsaturated fatty acids ($C_{18:1}$, $C_{18:2}$, $C_{18:3}$) in the generated soybean sprouts to varying degrees. For example, compared with those in the control, treatment with chlorothalonil reduced the contents of $C_{18:1}$, $C_{18:2}$, and $C_{18:3}$ in soybean sprouts by approximately 14%, 19%, and 27%, respectively (Table 3). Similarly, malathion treatment reduced the contents of $C_{18:1}$, $C_{18:2}$, and $C_{18:3}$ by approximately 27%, 31%, and 40%, respectively, compared with those in the control (Table 3). However, the increased content of saturated fatty acids ($C_{18:0}$) or decreased content of unsaturated fatty acids ($C_{18:1}$, $C_{18:2}$, and $C_{18:3}$) regulated by the pesticides in soybean sprouts was reversed by melatonin treatment (Table 3). For example, in soybean seeds treated with chlorothalonil and malathion, the application of melatonin reduced the $C_{18:0}$ content in soybean sprouts by approximately 12% and 19%, respectively, compared with that in the control (Table 3). Compared to those in the malathion treatment group, applying melatonin increased the contents of $C_{18:1}$, $C_{18:2}$, and $C_{18:3}$ in soybean sprouts by approximately 26%, 29%, and 51%, respectively (Table 3).

## Discussion

In the present study, two pesticides (chlorothalonil and malathion) were applied individually on soybean seeds, and high levels of pesticide residues in the treated soybean seeds were detected (Table 1). Compared with the contaminated seeds, the process of germination efficiently removed more than 95% of chlorothalonil and malathion in germinated soybean seeds after imbibition for 7 days. However, the high concentration of chlorothalonil (10 mM) and malathion (1 mM) in polluted soybean seeds resulted in a reduced germination rate (below to 85%) and sprout growth (about 90% of the normal growth) compared with those in the pesticide-free seeds. Seed germination requires an appropriate level of reactive oxygen species (ROS) [50], and pesticides, as "xenobiotics", might stimulate seed cells to produce excessive amounts of ROS [51], leading to a slight inhibitory effect on soybean seeds. Moreover, more than 7 days is too long to completely remove pesticide residues from contaminated soybean seeds. Published data shows that melatonin can improve seed germination and sprouting under stress conditions [26]. Herein, melatonin application to pesticide-polluted soybean seeds improved their germination and sprouting (Fig 2). The melatonin-enhanced germination ability of soybean seeds contaminated with pesticides might be related to its strong ROS scavenging ability [25]. Interestingly, melatonin could accelerate the degradation of the two pesticides, removing them completely within 5 days (Table 1). A similar study showed that exogenous melatonin could promote the degradation of pesticide residues in harvested jujube

fruits [24]. This suggested that germination, especially combined with melatonin, could be considered as a new biodegradation method to remove pesticide residues from contaminated soybean seeds. However, how does melatonin accelerate the degradation of pesticide residues in contaminated soybean seeds caused by the germination process?

Reports have shown that plants require the synergistic cooperation of glutathione and related enzymes (GR and GST) to complete the biodegradation of pesticide residues [21,22,52]. During seed germination, strong respiration leads to the production of a large amount of ROS, requiring the activation of enzymes such as GR to generate a large amount of reduced ascorbic acid and glutathione to alleviate the resultant oxidative stress [53]. Herein, in soybean sprouts, the GSH content and the related enzyme activities (GR and GST) were affected significantly by pesticides compared with those in the water control (Fig 3). This indicated that GSH and its related detoxifying enzymes produced during seed germination might be involved in the slow degradation of these pesticides [52]. Sufficient GSH synthesis is required to rapidly detoxify organic pesticides and to conjugate and excrete GSH-conjugated metabolites [52]. Moreover, melatonin can induce GR and GST activities and accelerate the degradation of pesticides in postharvest jujube [24]. Herein, the GSH content and enzyme activities (GR and GST) were enhanced significantly by melatonin (Fig 3). This is partly consistent with previous reports that indicated that plant melatonin and brassinosteroid-mediated pesticide degradation requires GSH and GST [23,24]. Based on these data, we suggested that the melatonin-induced increase in GSH production and GR/GST enzyme activities would contribute to the accelerated pesticide biodegradation in germinated soybean seeds and sprouts. However, this raised another question: how does melatonin affect the antioxidant nutrient accumulation in pesticide-treated soybean seeds after germination?

Compared with the control, treatment with pesticides drastically decreased ascorbic acid accumulation, but enhanced the total phenolics and isoflavone content, coupled with an enhanced total antioxidant capacity, in soybean sprouts after 7 d (Fig 4). This result partially agreed with that of a previous study, which reported the pesticide-mediated promotion of phenolics accumulation in soybean leaves [54]. Interestingly, these pesticide-mediated changes in antioxidant nutrient accumulation and antioxidant capacity were enhanced by melatonin in contaminated soybean seeds compared with those in the water control at 7 days post-germination (Fig 4). Soybean sprouts often contain high levels of bioactive phenolic metabolites (e.g., coumaric acid, ferulic acid, and naringin) and isoflavones (e.g., daidzin, glycidyl, genistein, daidzein, and genistein) [44]. In a subsequent study, it was found that soybean sprouts could accumulate more phenolic and isoflavone metabolites after melatonin treatment than soybean seeds (Table 2). This indicated that melatonin could significantly promote the accumulation of these secondary metabolites in soybean sprouts, especially isoflavones, which are beneficial to human health [55,56]. Moreover, the fatty acid composition has been considered a biomarker to determine the response of organisms to pesticides [57,58]. The decrease in the unsaturated fatty acid content ($C_{18:1}$, $C_{18:2}$, and $C_{18:3}$) in soybean sprouts caused by pesticides could be significantly restored by exogenous melatonin (Table 3). The decrease in the content of these three unsaturated fatty acids could be partially attributed to the oxidative stress caused by the pesticides, which leads to the consumption and degradation of these antioxidant substances [39]. Melatonin not only activates the antioxidant system of cells, but also has a strong ability to scavenge ROS [25]. Consequently, melatonin treatment can alleviate the oxidative stress caused by pesticides [51], thereby partially increasing the content of these unsaturated fatty acids (Table 3). Similar to the aforementioned antioxidants, unsaturated fatty acids have beneficial effects on human health because of their ability to scavenge free radicals [37–39]. Our research suggested that melatonin can improve the quality of soybean sprouts by increasing the accumulation of antioxidants and the content of unsaturated fatty acids. Biodegradation

methods are considered highly efficient approaches and are superior to physical methods because of their greater ability to remove organic toxins [59]. Thus, this melatonin-promoted seed germination and antioxidant nutrient production, coupled with pesticide residue removal, represents a new choice to produce safe and antioxidant-enriched food. However, this method still has some limitations. For example, we only tested two pesticides and one type of grain, thus it is unknown whether melatonin combined germination treatment would accelerate pesticide degradation and increase the accumulation of bioactive metabolites in other grains contaminated with other pesticides. In addition, the molecular mechanism by which melatonin promotes pesticide degradation in germinated seeds is unclear, the elucidation of which requires more experimental evidence.

## Conclusions

The use of pesticides as a last resort to prevent and control pests and diseases in grains during storage will inevitably lead to a large amount of pesticide residues on the grains. Herein, we developed a new method to avoid the toxicity caused by consuming these grains. Soybean seeds were first treated with chlorothalonil and malathion, followed by combined treatment with melatonin and germination. This not only rapidly removed the pesticide residues from the grains, but also increased the accumulation of various bioactive substances in the sprouted grains. The combined treatment significantly increased the overall content of vitamin C, polyphenols, and flavonoids, and markedly increased the accumulation of polyunsaturated fatty acids (such as $C_{18:1}$, $C_{18:2}$, and $C_{18:3}$). We hypothesized that the mechanism by which melatonin accelerates the removal of pesticide residues in grains and increases the accumulation of antioxidant nutrients is related to its ability to enhance glutathione synthesis and related enzyme activity in sprouted grains. Melatonin is a hormone that is easily obtainable, inexpensive, and essential for human health. Therefore, using melatonin combined with germination treatment represents a feasible method to obtain high-quality food from pesticide-contaminated grains.

## Supporting information

**S1 File.**
(ZIP)

## Acknowledgments

We would like to thank Dr. Benliang Deng for his comments in this work. We also thank the native English-speaking scientists from Elixigen Company (Huntington Beach, California) for editing our manuscript.

## Author Contributions

**Conceptualization:** Lingyun Li, Baoyan Li, Shan Tian.

**Data curation:** Henghua Qu, Zimeng Xu, Lulu Zhao, Xueqin Li.

**Formal analysis:** Henghua Qu, Zimeng Xu, Lulu Zhao, Xueqin Li.

**Funding acquisition:** Shan Tian, Baoyou Liu.

**Investigation:** Lingyun Li, Henghua Qu, Zimeng Xu, Lulu Zhao, Xueqin Li.

**Methodology:** Henghua Qu, Zimeng Xu, Lulu Zhao, Xueqin Li.

**Supervision:** Shan Tian, Baoyou Liu.

Writing – original draft: Baoyan Li, Shan Tian.

Writing – review & editing: Lingyun Li.

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
