## [Decision Letter · Decision Letter 0]

18 Mar 2024

PONE-D-23-43778A new method based on melatonin-mediated seed germination for quickly removing pesticide residues and improving nutritional quality for contaminated grainsPLOS ONE

Dear Dr. Tian,

Thank you for submitting your manuscript to PLOS ONE. After careful consideration, we feel that it has merit but does not fully meet PLOS ONE’s publication criteria as it currently stands. Therefore, we invite you to submit a revised version of the manuscript that addresses the points raised during the review process.

We look forward to receiving your revised manuscript.

Kind regards,

Rahul Kumar Tiwari, PhD

Academic Editor

PLOS ONE

A clean copy of the edited manuscript (uploaded as the new *manuscript* file)”.

3. PLOS requires an ORCID iD for the corresponding author in Editorial Manager on papers submitted after December 6th, 2016. Please ensure that you have an ORCID iD and that it is validated in Editorial Manager. To do this, go to ‘Update my Information’ (in the upper left-hand corner of the main menu), and click on the Fetch/Validate link next to the ORCID field. This will take you to the ORCID site and allow you to create a new iD or authenticate a pre-existing iD in Editorial Manager. Please see the following video for instructions on linking an ORCID iD to your Editorial Manager account: https://www.youtube.com/watch?v=_xcclfuvtxQ.

5. Please remove your figures from within your manuscript file, leaving only the individual TIFF/EPS image files, uploaded separately. These will be automatically included in the reviewers’ PDF.

Reviewers' comments:

Reviewer's Responses to Questions

**Comments to the Author**

1. Is the manuscript technically sound, and do the data support the conclusions?

Reviewer #1: Yes

Reviewer #2: No

2. Has the statistical analysis been performed appropriately and rigorously? 

Reviewer #1: Yes

Reviewer #2: Yes

3. Have the authors made all data underlying the findings in their manuscript fully available?

Reviewer #1: Yes

Reviewer #2: Yes

4. Is the manuscript presented in an intelligible fashion and written in standard English?

Reviewer #1: Yes

Reviewer #2: No

5. Review Comments to the Author

Reviewer #1: Dear authors,

After reviewing the paper entitled “A new method based on melatonin-mediated seed germination for quickly removing pesticide residues and improving nutritional quality for contaminated grains” by Li et al., a new method for removing pesticides from grains contaminated with pesticides has been developed using a combination of melatonin and germination. Not only can it effectively and quickly remove pesticide residues, but it can also increase the accumulation of antioxidants. The experimental design for this work is feasible, the data is reliable, and the necessity of the research work is clearly explained in the introduction. The significance and advantages and disadvantages of the experimental results were also discussed in depth. It is indeed a very interesting and meaningful work. However, there are also some minor problems that need to be improved.

1. Methods. It is recommended to use a flowchart for the experimental design section, which may be more conducive to understanding.

2. It is recommended to include references for the detection of vitamin C.

3. Figure 2 shows the experimental results of melatonin treatment for 24 hours, but the experimental design states that GSH, GR, and GST will be detected after 7 days of treatment. This may be due to the author's mistake or other reasons, which need to be clarified.

4. In the discussion section of L339-341, it is suggested to replace "combined treatment

with exogenous melatonin and pesticides" with "melatonin treatment", which emphasizes the promoting effect of melatonin.

5. At the end of the discussion section, it is suggested to provide suggestions for the shortcomings and improvement measures of this method, which would be better.

Best wishes

Your sincerely

Reviewer #2: The present manuscript is entitled “A new method based on melatonin-mediated seed germination for quickly removing pesticide residues and improving nutritional quality for contaminated grains”.

The presented manuscript is fascinating. This is extensive research, with a lot of analysis. A special contribution of the manuscript was focused on melatonin-mediated seed germination for quick removal of pesticide residues and improving nutritional quality. I found the research paper appropriate for publication in the Journal, but only after major modifications and clarification from the authors.

Please consider the text at the end of this letter

Keeping in mind the present form a few points which can be incorporated to enhance the quality and visibility of the manuscript

Major Comments

• The abstract portion is weak in a sense as not cover the whole idea of this research.

• The manuscript requires enhancement in its writing. It contains numerous superfluous sentences and grammatical errors that need to be addressed.

• Must write about the specification of the Gas chromatography instrument.

• Material and method are incomplete not mentioned about melatonin brand, preparation same applied for pesticides applied.

• Rather than writing this “ Here, an interesting question arises: why can the germination

• 313 process promote the degradation of pesticide residues in contaminated soybean seeds?” The author must explain the reason for this.

• Conclusion needs improvement in writing.

• Discussion is not more input.

Minor Comments

• Must check the spelling mistakes in the whole manuscript like line no. 26 “germination”.

6. PLOS authors have the option to publish the peer review history of their article (what does this mean?). If published, this will include your full peer review and any attached files.

Reviewer #1: No

Reviewer #2: **Yes: **Chirag Maheshwari

---

## [Author Response · Author response to Decision Letter 0]

2 Apr 2024

Response to comments

Dear Editor,

I have received the comments for the manuscript entitled “A new method based on melatonin-mediated seed germination to quickly remove pesticide residues and improve the nutritional quality of contaminated grains”(Ms. No. PONE-D-23-43778). I am very grateful for the advice received from the reviewers and the editor. Several major errors have been corrected and they are listed below. I hope that this paper can now be accepted and published in your journal Plos one.

Best wishes

You sincerely 

Shan Tian, Ph.D.

PONE-D-23-43778

A new method based on melatonin-mediated seed germination for quickly removing pesticide residues and improving nutritional quality for contaminated grains

PLOS ONE

Dear Dr. Tian,

Thank you for submitting your manuscript to PLOS ONE. After careful consideration, we feel that it has merit but does not fully meet PLOS ONE’s publication criteria as it currently stands. Therefore, we invite you to submit a revised version of the manuscript that addresses the points raised during the review process.

We look forward to receiving your revised manuscript.

Kind regards,

Rahul Kumar Tiwari, PhD

Academic Editor

PLOS ONE

A clean copy of the edited manuscript (uploaded as the new *manuscript* file)”.

3. PLOS requires an ORCID iD for the corresponding author in Editorial Manager on papers submitted after December 6th, 2016. Please ensure that you have an ORCID iD and that it is validated in Editorial Manager. To do this, go to ‘Update my Information’ (in the upper left-hand corner of the main menu), and click on the Fetch/Validate link next to the ORCID field. This will take you to the ORCID site and allow you to create a new iD or authenticate a pre-existing iD in Editorial Manager. Please see the following video for instructions on linking an ORCID iD to your Editorial Manager account: https://www.youtube.com/watch?v=_xcclfuvtxQ.

5. Please remove your figures from within your manuscript file, leaving only the individual TIFF/EPS image files, uploaded separately. These will be automatically included in the reviewers’ PDF.

Reviewers' comments:

Reviewer's Responses to Questions

Comments to the Author

1. Is the manuscript technically sound, and do the data support the conclusions?

Reviewer #1: Yes

Reviewer #2: No

2. Has the statistical analysis been performed appropriately and rigorously?

Reviewer #1: Yes

Reviewer #2: Yes

3. Have the authors made all data underlying the findings in their manuscript fully available?

Reviewer #1: Yes

Reviewer #2: Yes

4. Is the manuscript presented in an intelligible fashion and written in standard English?

Reviewer #1: Yes

Reviewer #2: No

5. Review Comments to the Author

Reviewer #1: Dear authors,

After reviewing the paper entitled “A new method based on melatonin-mediated seed germination for quickly removing pesticide residues and improving nutritional quality for contaminated grains” by Li et al., a new method for removing pesticides from grains contaminated with pesticides has been developed using a combination of melatonin and germination. Not only can it effectively and quickly remove pesticide residues, but it can also increase the accumulation of antioxidants. The experimental design for this work is feasible, the data is reliable, and the necessity of the research work is clearly explained in the introduction. The significance and advantages and disadvantages of the experimental results were also discussed in depth. It is indeed a very interesting and meaningful work. However, there are also some minor problems that need to be improved.

1. Methods. It is recommended to use a flowchart for the experimental design section, which may be more conducive to understanding.

Response: Thank you for this suggestion. We have added the flowchart.

2. It is recommended to include references for the detection of vitamin C.

Response: Thank you. We have added the references.

3. Figure 2 shows the experimental results of melatonin treatment for 24 hours, but the experimental design states that GSH, GR, and GST will be detected after 7 days of treatment. This may be due to the author's mistake or other reasons, which need to be clarified.

Response: The results measured here indeed reflect the GSH and related enzyme activities after 24 hours of melatonin treatment. Perhaps it was because the experimental methods were not clearly written, leading to misunderstanding. In fact, on the 5th day after melatonin treatment, both pesticide residues disappeared and enzyme activity decreased. Therefore, the higher enzyme activity in Fig. 2 was the result of the data after 24 hours of treatment, not on the 7th day. To avoid misunderstanding, we have revised the experimental section.

4. In the discussion section of L339-341, it is suggested to replace "combined treatment

with exogenous melatonin and pesticides" with "melatonin treatment", which emphasizes the promoting effect of melatonin.

Response: Thank you. Pesticide stimulation can also slightly increase the accumulation of secondary metabolites. However, to emphasize the stimulating effect of melatonin rather than pesticides, we have made revisions according to the reviewer's comments. 

5. At the end of the discussion section, it is suggested to provide suggestions for the shortcomings and improvement measures of this method, which would be better.

Response: Thank you for this comment. We have made the requested revisions.

Best wishes

Your sincerely

Reviewer #2: The present manuscript is entitled “A new method based on melatonin-mediated seed germination for quickly removing pesticide residues and improving nutritional quality for contaminated grains”.

The presented manuscript is fascinating. This is extensive research, with a lot of analysis. A special contribution of the manuscript was focused on melatonin-mediated seed germination for quick removal of pesticide residues and improving nutritional quality. I found the research paper appropriate for publication in the Journal, but only after major modifications and clarification from the authors.

Please consider the text at the end of this letter

Keeping in mind the present form a few points which can be incorporated to enhance the quality and visibility of the manuscript

Major Comments

• The abstract portion is weak in a sense as not cover the whole idea of this research.

Response: Thank you, we have improved the abstract.

• The manuscript requires enhancement in its writing. It contains numerous superfluous sentences and grammatical errors that need to be addressed.

Response: Thank you for this comment. The manuscript has been revised by an native English-speaking editor.

• Must write about the specification of the Gas chromatography instrument.

Response: Thank you for this comment. We have added the specification of the Gas chromatography instrument.

• Material and method are incomplete not mentioned about melatonin brand, preparation same applied for pesticides applied.

Response: Thank you for this comment. The relevant text has been improved

• Rather than writing this “ Here, an interesting question arises: why can the germination process promote the degradation of pesticide residues in contaminated soybean seeds?” The author must explain the reason for this.

Response: Thank you for this comment. We made a small correction. Our original intention was to explain how melatonin can accelerate the degradation of pesticides in seeds mediated by germination.

• Conclusion needs improvement in writing.

Response: Thank you for this comment. The conclusion section has been improved.

• Discussion is not more input.

Response: Thank you, the discussion has been improved.

Minor Comments

• Must check the spelling mistakes in the whole manuscript like line no. 26 “germination”.

Response: The manuscript has been checked carefully for spelling mistakes; however, in the example you quote, the word is spelled correctly.

6. PLOS authors have the option to publish the peer review history of their article (what does this mean?). If published, this will include your full peer review and any attached files.

Do you want your identity to be public for this peer review? For information about this choice, including consent withdrawal, please see our Privacy Policy.

Reviewer #1: No

Reviewer #2: Yes: Chirag Maheshwari

In addition, we have made corrections to some minor errors found in the paper. For example, the concentration of malathion should be 1 mM instead of 0.1 mM (which has been corrected); We added a melatonin treatment group to the seed germination rate and shoot length in Figure 2, and changed the time to 1, 3, 5, and 7 days, and adjusted it to a bar chart. Finally, the major changes are shown in RED in the revised manuscript. We would like to express our gratitude to the reviewers and editors for their valuable comments on this article.

Best wishes

Your sincerely

Shan Tian, Ph.D.

---

## [Decision Letter · Decision Letter 1]

18 Apr 2024

A new method based on melatonin-mediated seed germination to quickly remove pesticide residues and improve the nutritional quality of contaminated grains

PONE-D-23-43778R1

Dear Dr. Tian,

We’re pleased to inform you that your manuscript has been judged scientifically suitable for publication and will be formally accepted for publication once it meets all outstanding technical requirements.

Kind regards,

Rahul Kumar Tiwari, PhD

Academic Editor

PLOS ONE

Additional Editor Comments (optional):

Reviewers' comments:

Reviewer's Responses to Questions

**Comments to the Author**

1. If the authors have adequately addressed your comments raised in a previous round of review and you feel that this manuscript is now acceptable for publication, you may indicate that here to bypass the “Comments to the Author” section, enter your conflict of interest statement in the “Confidential to Editor” section, and submit your "Accept" recommendation.

Reviewer #1: All comments have been addressed

Reviewer #2: All comments have been addressed

2. Is the manuscript technically sound, and do the data support the conclusions?

Reviewer #1: Yes

Reviewer #2: Yes

3. Has the statistical analysis been performed appropriately and rigorously? 

Reviewer #1: Yes

Reviewer #2: Yes

4. Have the authors made all data underlying the findings in their manuscript fully available?

Reviewer #1: Yes

Reviewer #2: Yes

5. Is the manuscript presented in an intelligible fashion and written in standard English?

Reviewer #1: Yes

Reviewer #2: Yes

6. Review Comments to the Author

Reviewer #1: The author made necessary revisions based on the review comments. This is indeed a very creative and persuasive job. I fully agree that the current status of the paper can be accepted and published in PLOS ONE.

Reviewer #2: No additional comments are necessary as all previous comments have been appropriately addressed to satisfaction.

7. PLOS authors have the option to publish the peer review history of their article (what does this mean?). If published, this will include your full peer review and any attached files.

Reviewer #1: No

Reviewer #2: No
